# Experimental Verification of Contact Acoustic Nonlinearity at Rough Contact Interfaces

**DOI:** 10.3390/ma14112988

**Published:** 2021-05-31

**Authors:** Youngbeom Kim, Sungho Choi, Kyung-Young Jhang, Taehyeon Kim

**Affiliations:** 1Department of Mechanical Convergence Engineering, Hanyang University, 222 Wangsimni-ro, Seongdong-gu, Seoul 04763, Korea; ybkim0209@hanyang.ac.kr; 2LANL-JBNU Engineering Institute-Korea, Jeonbuk National University, Jeonju-si 54896, Jeollabuk-do, Korea; 3School of Mechanical Engineering, Hanyang University, Seoul 04763, Korea; 4Radiation and Decommissioning Laboratory, KHNP-CRI, Daejeon 34101, Korea; taehyeon.kim@khnp.co.kr

**Keywords:** interfacial stiffness, contact acoustic nonlinearity (CAN), ultrasonic, contact condition, NDT, longitudinal wave, roughness, Al6061-t6, nonlinear ultrasonics

## Abstract

When a longitudinal wave passes through a contact interface, second harmonic components are generated due to contact acoustic nonlinearity (CAN). The magnitude of the generated second harmonic is related to the contact state of the interface, of which a model has been developed using linear and nonlinear interfacial stiffness. However, this model has not been sufficiently verified experimentally for the case where the interface has a rough surface. The present study verifies this model through experiments using rough interfaces. To do this, four sets of specimens with different interface roughness values (Ra = 0.179 to 4.524 μm) were tested; one set consists of two Al6061-T6 blocks facing each other. The second harmonic component of the transmitted signal was analyzed while pressing on both sides of the specimen set to change the contact state of the interface. The experimental results showed good agreement with the theoretical prediction on the rough interface. The magnitude of the second harmonic was maximized at a specific contact pressure. As the roughness of the contact surface increased, the second harmonic was maximized at a higher contact pressure. The location of this maximal point was consistent between experiments and theory. In this study, an FEM simulation was conducted in parallel and showed good agreement with the theoretical results. Thus, the developed FEM model allows parametric studies on various states of contact interfaces.

## 1. Introduction

With development of nuclear, aviation, and power plant industries, which require high reliability and safety, the importance of flaw detection is increasing for safety diagnosis and integrity evaluation of structures. For this, ultrasonic inspection has been widely used; however, it is difficult for conventional ultrasonic flaw detection technologies to detect partially closed micro-scale defects caused by stress corrosion or thermal fatigue for which the crack surface has formed a contact interface due to thermal expansion or external stress. This limitation is because conventional methods are based on linear wave propagation and mostly use the amplitude change of ultrasonic waves reflected at or transmitted through the defect surface. However, the amplitude change at a closed interface is not significant and is difficult to detect. To solve this problem, nonlinear ultrasonic methods based on contact acoustic nonlinearity (CAN) have been studied [1].

CAN is a phenomenon in which harmonic waves are generated due to a temporary opening and closing of the interface or a nonlinear pressure–displacement relationship when ultrasonic waves are reflected at or transmitted through the contact interface [2,3,4,5]. Related theories have been studied for decades. Richardson et al. [6] analyzed the nonlinear dynamics of a system composed of an unbonded planar interface separating two semi-infinite linear elastic media. This is referred to as the hard contact condition, where the opening and closing of the interface is the origin of the nonlinearity. However, this theory does not consider that the contact state is variable with surface state. To supplement this point, Rudenko et al. [7] studied the soft contact condition with the distributed-microasperity model.

Later, the ultrasonic response was quantified by modeling the contact interface with a rough surface as a spring with linear and nonlinear contact stiffness [8,9,10], where the contact stiffness is dependent on the static pressure applied at the interface. This was followed by experimental verification by several researchers. Drinkwater et al. [11] put in contact two aluminum block specimens, applied pressure from both sides, and analyzed the reflected wave at different excitation frequencies (4 MHz to 17 MHz) to confirm the frequency dependence and the relationship between the reflectivity and the contact stiffness by increasing the static pressure. Nam et al. [12] carried out similar experiments but used an ultrasonic wave that was obliquely incident on the contact interface. Furthermore, Biwa et al. [13,14] tested the contact interface by pressing together two aluminum blocks with surface roughness and measured the contact stiffness against static pressure at the interface with different roughness values. The dependence of the second-order harmonic amplitude on the incident wave amplitude was verified experimentally only on a relatively smooth contact interface (the surface roughness Ra ≤ 1 μm). 

Therefore, in this study, we tried to confirm experimentally whether the CAN theory can be applied to a rough contact interface. To do this, four sets of two aluminum blocks with different surface roughness values (Ra = 0.179 to 4.524 μm) at the contact face were prepared. Two aluminum block specimens were put in contact, and static pressure was applied from both sides. The transmitting transducer was placed on one of the pressing surfaces, and the receiving transducer was placed on the other to receive the ultrasonic waves transmitted through the contact interface. While increasing the static pressure to 80 MPa, the change in transmission efficiency was measured to estimate the linear and nonlinear interfacial stiffness values expressed as a power function of pressure. From this, the amplitude of the second harmonic generated by CAN was obtained based on Biwa’s theory and compared with the experimental results. Since the contact interface of an actual crack, such as a stress corrosion crack or fatigue crack, can be very rough, this study will be useful by confirming that the CAN theory can be applied to detect actual closed cracks.

Additionally, second harmonics can be generated by inherent material nonlinearity or by measurement system nonlinearity [15]. Therefore, a theoretical prediction that does not consider those extra harmonic components will show a difference from the experimental results. In this study, such a difference was verified by conducting a separate experiment for conditions without the CAN effect (using a single specimen without an interface), showing that the difference was due to nonlinearities other than those caused by the CAN effect.

In addition, since it is difficult to experimentally test the contact interface in various contact states, a numerical analysis approach that can replace the experiment will be useful. Therefore, in this study, numerical analysis using the finite element method (FEM) was performed, and it was determined whether the results fit well with the theoretical results.

The remainder of this paper is organized as follows. Section 2 presents a brief description of the CAN theory with the theoretically estimated transmission efficiency and second harmonic amplitude according to variations of contact pressure. Section 3 describes the FEM modeling and simulation results. Section 4 describes the fabrication process used to create test samples, as well as the experimental setup. It also compares our experimental results with the results of the theoretical prediction and FEM simulation. Section 5 presents our conclusions, the limitations of the current study, and suggestions for future works.

## 2. Contact Acoustic Nonlinearity at a Contact Interface

### 2.1. Theory

The longitudinal wave propagation through a contact interface with roughness can be analyzed using a spring model, as shown in Figure 1. Since the details of this theory are well described in the literature [5], it is only briefly introduced here. When pressure is applied to such an interface, large asperities collide and deform, producing elastic and plastic deformation. Therefore, the pressure–displacement relationship becomes nonlinear and can be expressed as follows:(1)$$P\left(h\right){=P}_{0}-K_{1}\cdot \left({h\;-\;h}_{0}\right){+K}_{2}\cdot {{(h\;-\;h}_{0})}^{2}$$

Here, *P* and *h* are the pressure and gap displacement, respectively. *P*_0_ is the static contact pressure, and h_0_ is the initial gap at *P*_0_, i.e., *P*_0_ = *P* (*h*_0_). *K*_1_ and *K*_2_ are the linear and nonlinear interfacial stiffness, respectively, and can be defined as follows from Equation (1).
(2)$$K_{1}=-\;(\frac{\partial \left(P\right)}{\partial h}{)}_{{h=h}_{0}}{\;,\;K}_{2}=\frac{1}{2}{\left(\frac{\partial^{2}\left(P\right)}{\partial h^{2}}\right)}_{{h=h}_{0}}$$

Considering one-dimensional propagation, the approximate solution of the transmitted wave can be obtained as displacement for a harmonic wave as follows [16].
(3)$$U_{T}\left(t\;\right)=\frac{K_{2}A_{0}{}^{2}}{K_{1}\left\{{1+4K}_{1}{}^{2}{/(\rho }^{2}c^{2}\omega^{2})\right\}}+\frac{{2K}_{1}A_{0}}{\rho c\omega \sqrt{{1+4K}_{1}{}^{2}{/(\rho c\omega )}^{2}}}\cos\left({\omega t-\delta }_{1}\right)-\frac{K_{2}A^{2}{}_{0}}{\rho c\omega \left\{{1+4K}_{1}{}^{2}{/(\rho c\omega )}^{2}\right\}\sqrt{{1+K}_{1}{}^{2}{/(\rho c\omega )}^{2}}}\;{\times \sin(2\omega t\;-\;2\delta }_{1}{+\delta }_{2})$$

Here, *A*_0_ is the displacement amplitude of the incident wave, c is the longitudinal wave velocity, ρ is the density of the material, ω is the angular frequency of the incident wave, $\delta_{1}=tan^{-1}\left(\frac{\rho c\omega }{2K_{1}}\right)$, and $\delta_{2}=$
$tan^{-1}\left(\frac{K_{1}}{\rho c\omega }\right)$. The first term on the right side is the static displacement component. The second term represents the fundamental component of the incident wave frequency, which is a linear response, and its amplitude depends only on linear stiffness. The third term indicates the second harmonic component with a frequency twice that of the incident wave, and its amplitude depends not only on linear stiffness *K*_1_, but also on the nonlinear stiffness *K*_2_. 

Then, amplitude *A*_1_ of the fundamental component and amplitude *A*_2_ of the second harmonic component in the transmitted wave can be determined from Equation (3) as follows.
(4)$$A_{1}=\frac{{2K}_{1}A_{0}}{\rho c\omega \sqrt{1+\frac{{4K}_{1}{}^{2}}{{\left(\rho c\omega \right)}^{2}}}}$$
(5)$$A_{2}=\frac{{(K}_{2}A_{0}A_{1})}{{2K}_{1}\sqrt{1+\frac{{4K}_{1}{}^{2}}{{\left(\rho c\omega \right)}^{2}}}\sqrt{1+\frac{K_{1}{}^{2}}{{\left(\rho c\omega \right)}^{2}}}}$$

Note that some terms of the amplitude of the harmonic component have been replaced by *A*_1_ in Equation (5).

Meanwhile, the transmission efficiency is defined as the ratio of *A*_1_ and *A*_0_, as follows.
(6)$$T=\frac{{2K}_{1}}{\rho c\omega \sqrt{1+\frac{{4K}_{1}{}^{2}}{{\left(\rho c\omega \right)}^{2}}}}$$

As linear stiffness *K*_1_ increases, transmission efficiency increases. Therefore, as linear stiffness increases, the interface closes further. When *K*_1_ increases to much greater than ρcω, the transmission efficiency converges to 1, and linear stiffness *K*_1_ can be expressed as a power function of static pressure $P_{0}$, as follows [5,13,14].
(7)$$K_{1\;}{=CP}_{0}{}^{m}$$

Here, *C* and *m* are positive constants related to the roughness of the interfacial surface. Substituting Equation (7) into Equation (6), transmission efficiency T is expressed as a function of pressure $P_{0}$. Then, the constants *C* and *m* can be determined by fitting the experimental results of T measured while varying static pressure $P_{0}$. Furthermore, the nonlinear stiffness *K*_2_ of Equation (2) can be expressed as follows.
(8)$$K_{2}=\frac{1}{2}K_{1}\frac{{dK}_{1}}{{dP}_{0}}=\frac{1}{2}{mC}^{2}P_{0}{}^{2m-1}$$

Therefore, Equation (1) can be rewritten as
(9)$${P=P}_{0}-{CP}_{0}{}^{m}\left({h\;-\;h}_{0}\right)+\frac{1}{2}{mC}^{2}p_{0}{}^{2m-1}{{(h\;-\;h}_{0})}^{2}$$

### 2.2. Theoretical Simulations of A_1_ and A_2_

A theoretical simulation for the specimen to be tested in the experiment was conducted using the aforementioned theoretical model. In the experiment (Section 4.1), four sets of specimens in which two aluminum alloy (Al6061-T6) blocks were contacted were tested, and the surface roughness of each set was varied. The interfacial surface roughness values of the specimens were Ra = 0.179, 1.458, 2.567, and 4.524 μm, respectively (Table 1). *K*_1_ and *K*_2_ were obtained using the constants C and m, which were determined in the experiment (Table 2) to calculate *A*_1_ and *A*_2_. Other parameters used for simulation were $\rho =2700\;\mathrm{kg}/m^{3},$ c = 6300 m/s, and frequency = 2 MHz, as in the experiment. The results are shown in Figure 2, where the amplitudes were normalized to the maximum value.

When the contact pressure is small, since the interface is open, the amplitude of fundamental component *A*_1_ is almost zero. As the pressure increases, the interface gradually closes and *A*_1_ increases. With a rougher interface, *A*_1_ starts to rise at a higher pressure. The normalized value of *A*_1_ converges to 1, which corresponds to a closed interface state, and the two specimens are considered as one body with no interface. Additionally, the rougher the interface, the greater the contact pressure when transmittance approaches 1. For second harmonic amplitude *A*_2_, a peak appears at a specific pressure, and the rougher the interface, the greater the peak pressure. This means that the CAN effect can be maximized at an appropriate interfacial gap. Additionally, the rougher the interface, the higher the pressure needed to achieve the appropriate gap.

## 3. Finite Elements Method

### 3.1. D Model for Longitudinal Wave Propagation through a Contact Interface

Finite element analysis is carried out using the ABAQUS tool to verify the theoretical model to analyze structural nonlinear contact problems, and can easily handle various contact conditions by simply changing the contact properties. Figure 3 shows the two-dimensional model of two aluminum blocks in contact with an interface, on the side of the model, a symmetry constraint is applied to the plane of constant y coordinates. U1 and U2 are the degrees of freedom of translational motion along x and y axes, respectively, and UR1 is the degree of freedom of rotational motion due to the rotational moment about the x-axis. At the lower end, the displacement in the x-direction U1 is given as zero as a boundary condition, which means that the wave is 100% reflected at the lower end. This FEM modeling is based on plane-strain condition, and applied element shape is quadrilateral, element size is 0.1 mm, and the number of elements is 240,000. A longitudinal wave is input at the top as the oscillating displacement and propagated in the x-direction. The material properties are those of Al6061-T6: the density is $2700\;\mathrm{kg}/m^{3}$, Young’s modulus is $68\times 10^{9}\;N/m^{2}$, and Poisson’s ratio is 0.33. The input signal was a 2 MHz tone burst sine wave of eight cycles with a displacement amplitude of 10 nm.

To express the contact condition, the relation of the pressure gap shown in Equation (9) is used. In ABAQUS, however, a positive gap is referred to as clearance, and a negative gap is referred to as overclosure. Thus, the contact property at the interface can be defined as the pressure–overclosure relationship. In our analysis, Equation (10) was used instead of Equation (9), in which the gap term expressed as (*h* − *h*_0_) in Equation (9) was replaced with (−*h*), which expresses the overclosure amount.
(10)$$p\left(h\right){=p}_{0}{+CP}_{0}{}^{m}h+\frac{1}{2}{mC}^{2}P_{0}{}^{2m-1}h^{2}$$

Then, using constants *C* and *m* (Table 2) for each specimen obtained in the experiment, the contact condition can be defined using the above equation. Static uniform pressure from 0 to 80 MPa was applied to the top surface as compression stress. The signal transmitted across the contact interface was received at a center point on the lower end. The received signal was processed by MATLAB (MathWorks, Natick, MA, USA, vR2019b) using fast Fourier transform (FFT) with the Hanning window at a sampling frequency of 1 GHz.

### 3.2. Simulation Results

The simulation was conducted by changing the contact properties according to the surface roughness of the specimen. Figure 4 shows the simulation results for the contact state of specimen 2 and plots the received signals and their FFT spectra at three contact pressures of 0.1, 22.5, and 80 MPa. At a contact pressure of 0.1 MPa, the interface is in an almost open state and the signal amplitude is very weak. As the pressure increases, the signal amplitude gradually increases as the interface closes. The amplitude of fundamental component *A*_1_ and the amplitude of second harmonic *A*_2_ are determined from the magnitude of the spectrum at 2 MHz and 4 MHz, respectively. The second harmonic amplitude is greater at 22.5 MPa than at 80 MPa, although the fundamental amplitude at 22.5 MPa is smaller than that at 80 MPa.

Figure 5 shows the values of *A*_1_ and *A*_2_ for all specimens with respect to the applied static pressure, in which they were normalized to the maximum value. These results are similar to the theoretical calculations shown in Figure 2. Likewise, the rougher the surface, the higher the applied pressure needed to transmit the ultrasonic waves through the contact interface. The peak of the second harmonic occurs at higher contact pressure. A detailed comparison is described later with the experimental results.

## 4. CAN Experiment with Different Roughness Values

### 4.1. Specimens

Figure 6 shows the specimens used in the experiment. The experiment was carried out with four sets of contact interface roughness values. One set consists of two Al6061-T6 blocks facing each other to form one contact interface. The blocks are cylindrical, and each is 40 mm in diameter and 30 mm long. The contact surfaces of the two blocks in contact in a set have the same roughness.

The average roughness of the four specimen sets was measured using an optical microscope, and the roughness value (Ra) of each specimen set is shown in Table 1. Figure 7 shows the surface 3D images of the optical microscope.

**Table 1 materials-14-02988-t001:** Roughness of four specimen sets.

	Ra [μm]
Specimen 1	0.179
Specimen 2	1.458
Specimen 3	2.567
Specimen 4	4.524

### 4.2. Experimental Setup

The experimental setup is shown in Figure 8. A PZT transducer with a center frequency of 2.25 MHz was attached to the top surface of the upper block and used to transmit a 2 MHz longitudinal wave, and a PZT transducer with a center frequency of 5 MHz was attached to the bottom of the lower specimen and used to receive the second harmonic of the ultrasonic wave transmitted through the specimen. To align the axis of wave propagation, a specially designed jig was installed. To apply variable static pressure to the contact interface, a load of 0 to 10,000 kg was applied to both sides of the specimen using a hydraulic press; this range corresponds to contact pressures of about 0 to exactly 78.1 MPa. The input signal entering the transmitting transducer is 13 cycles of a tone-burst signal of 2 MHz, which was excited in a high-power tone-burst signal generator (RAM-5000, RITEC, Warwick, RI, USA), and data were acquired through a digital oscilloscope.

Figure 9 shows an example of the received signal for specimen 2 at 22.5 MPa. The time duration of the tone burst signal we used is about 7 μs, and considering that the time of flight for round trip between the interface and the lower boundary is about 10 μs, overlapping with multiple reflection signals can be sufficiently avoided. The acquired data were analyzed via FFT through MATLAB (MathWorks, Natick, MA, USA, vR2019b). Only seven cycles of the central part of the signal, where it was stable, were used for the FFT analysis, as shown in the figure. In addition, the Hanning window was applied to reduce the influence of side lobes. Figure 10 shows the FFT results of specimen 2 at three static pressures. The second harmonic components at 4 MHz were detected clearly at contact pressures of 22.5 and 80 MPa. The magnitude of the second harmonic component at 22.5 MPa was greater than that at 80 MPa. In contrast, the magnitude of the fundamental component was greater at 80 MPa, which is consistent with the results of the FEM analysis. 

### 4.3. Experimental Results

In the experiment, the pressure applied at the contact interface was increased, transmitted amplitude *A*_1_ was detected, and transmission efficiency T was estimated with respect to contact pressure to estimate linear stiffness. Figure 11 shows an example of the results for specimen 1, where the transmission efficiency was obtained by normalizing *A*_1_ to the maximum value obtained at the maximum pressure. The transmission efficiency converged to 1 as the pressure increased. The linear stiffness *K*_1_ against the contact pressure was calculated using Equation (6) on a point-by-point basis from the measured transmission efficiency, and the constants *C* and *m* were obtained by power-law fitting this data to Equation (7). A commercial software (Origin) was used for the nonlinear curve fitting. Note that the fitting range was set up to the contact pressure at which the transmission efficiency is sufficiently converged. In the case of specimen 1, the fitting range was up to 48 MPa, at which the transmission efficiency is 99%. The reason for this setting is as follows. What we pay attention to in this study is the variation in the amplitude of the harmonic component for the variation in the soft contact state, and such soft contact state is created before the transmission efficiency reaches 100%. However, after the upper limit of the set fitting range, the linear stiffness rises rapidly, so fitting over the entire pressure range can overfit the rapid-rising section and distort the analysis of the soft contact state. Therefore, we set the fitting range up to the pressure at which the transmission efficiency reaches 99%. This type of fitting was applied similarly to other specimens.

Figure 12 shows the experimentally obtained transmission efficiency and linear stiffness for all specimens with power-law fitting. When calculating the transmission efficiency and fitting the results, the abovementioned method was applied. Particularly, in specimens 3 and 4, the transmission efficiency did not sufficiently converge even at the applied maximum pressure, so the fitting range was set to the entire pressure range. The linear stiffness of specimen 1 (with a roughness of 0.179 μm) increased from a contact pressure of 0 MPa, while the linear stiffness of specimen 4 (with a roughness of 4.524 μm) started to increase at much higher pressure. In addition, in specimens 2, 3, and 4, we can see a weak up-and-down in the transmission efficiency at low pressure, although ideally the transmission efficiency should be close to zero there. This is probably due to the instability of experimental setup at low pressure. Nevertheless, as shown in Figure 12b, the stiffness at low pressure is very small, so the effect on analyzing the overall trend is negligible. However, since the linear stiffness data obtained in such up-and-down section is worthless for curve-fitting, it is considered more preferable not to include this section. Therefore, the lower limit of the curve fitting range was taken as the pressure at which the transmission efficiency starts to increase stably.

Table 2 shows the constant values obtained from all specimens. The constant *m* tends to increase as surface roughness increases, while the constant *C* tends to decrease on the contrary. The parameter standard errors for *m* and *C* were evaluated to be smaller than the fitted values, and the correlation coefficient R^2^ was also close to 1 in all specimens, which indicates that the fitting was done properly.

**Table 2 materials-14-02988-t002:** Constants C and m obtained for all specimens.

Sample	*m*	$C\;[\mathrm{MPa}{\;}^{1-m}\cdot {\mathrm{nm}}^{-1}]$	Standard Error. *m*	Standard Error. *C*	R^2^
Specimen 1	0.930	$2.41\times 10^{-2}$	0.016	1.32$\times 10^{-3}$	0.995
Specimen 2	2.601	4.09 $\times 10^{-5}$	0.075	$1.11\times 10^{-5}$	0.985
Specimen 3	4.987	$3.47\times 10^{-10}$	0.112	$1.61\times 10^{-10}$	0.988
Specimen 4	7.486	$5.85\times 10^{-15}$	0.147	$3.67\times 10^{-15}$	0.993

### 4.4. Comparison of Theoretical, FEM, and Experimental Results for A_1_ and A_2_

Finally, the theoretical simulations, FEM simulations, and experimental results were compared. Figure 13 and Figure 14 show the results for *A*_1_ and *A*_2_, respectively. For *A*_1_, the theoretical, FEM, and experimental results agree well with each other. In particular, the FEM results are in perfect agreement with the theoretical results. Experimental results show perfect agreement at high pressure but a slight difference at low pressure. This difference in the low-pressure region is because, as mentioned earlier, the interface did not involve sufficient contact. For *A*_2_, the peak pressure is consistent between theoretical, FEM, and experimental results. The tendency to decrease after passing the peak is the same. However, while the theoretical and FEM results continue to decrease, the experimental results show convergence at some level. The reasons for this result are discussed in the next section.

### 4.5. Difference between Experiment and Theory at High Contact Pressure

We only considered the contact acoustic nonlinear effect at the interface when conducting theoretical and FEM analysis. However, since the material nonlinear effect inherently exists regardless of the presence or absence of an interface, it is necessary to further examine this effect, although it is known that the magnitude of the harmonics caused by the material nonlinear effect is relatively small compared to the influence of the CAN effect. In addition, system-induced harmonics can be included in the received signal. At high pressures, the transmission efficiency is almost constant; therefore, the aforementioned extra second harmonics will be constant.

To confirm this, the magnitude of the second harmonic component in a single cylindrical Al6061-T6 specimen (60 mm in length) was measured. There was no interface in this specimen. To create the same propagation distance as in the previous experiment, the length of the specimen was made equal to the combined length of the two blocks. Since all experimental equipment and materials were the same as in the previous experiment, the material nonlinearity or the magnitude of the second harmonic component caused by the system was the same. The intention of this experiment was to determine whether the magnitude of *A*_2_ at high pressure (e.g., 80 MPa) shown in specimen 1 of Figure 14 is the same as that of *A*_2_ obtained from the specimen without an interface. For this comparison, we measured the relative parameter β′, which is defined as follows [17].
(11)$$\beta^{\prime }=\frac{A_{2}}{A_{1}{}^{2}}$$

Compare β′ calculated from the *A*_1_ and *A*_2_ values previously obtained for specimen 1 (with an interface) at a pressure of 80 MPa and β′ newly measured from the specimen without an interface. The average values of 10 repeated measurements were 0.0109 ± 0.0001 for specimen 1 at 80 MPa and 0.0106 ± 0.0002 for the single specimen. The results are similar to each other, which means that the convergence of *A*_2_ to a certain constant value at high pressure in the previous experiment was due to the extra harmonic component from material nonlinearity and system nonlinearity. Note that if the material nonlinearity and the system nonlinearity can be identified in advance, it will be possible to integrate the CAN model with the contribution of material nonlinearity and system nonlinearity. If such an integrated model is applied, it is expected that the problem of the difference between the theoretical prediction and the experimental results at high pressure shown above will be solved. However, although the model for material nonlinearity is well-known, the model for system nonlinearity is still a challenging task [15].

## 5. Conclusions

In this paper, the theory of contact acoustic nonlinearity at an interface with roughness, in which the contact state of the interface is represented using linear and nonlinear interfacial stiffness (which vary according to contact pressure), was experimentally verified. To do this, four sets of specimens with different interface roughness values (Ra = 0.179 to 4.524 μm) were tested. One set of specimens consisted of two AL6061-T6 blocks facing each other. The second harmonic component of the transmitted signal was analyzed while applying force to both sides of the specimen set to change the contact state of the interface. The experimental results showed good agreement with the theoretical prediction, even with a rough interface. The amplitude of the second harmonic component was maximized at a specific contact pressure. Additionally, as the roughness of the contact surface increased, the second harmonic component was maximized at a higher contact pressure. The location of this maximal point was consistent in experimental and theoretical results. At high pressure, however, the interfacial stiffness was very large so that the amplitude of the second harmonic converged to zero in the theoretical results, while maintaining a constant value in the experimental results. Based on additional experiments conducted on a single specimen with the same propagation distance but no interface, this difference was determined to be caused by the nonlinearity of the material itself and the component due to the system nonlinearity.

Additionally, FEM simulations were conducted in parallel, in which the contact interface was modeled in two dimensions using ABAQUS. Numerical simulation results for tested specimens were in good agreement with the theoretical predictions. The developed FEM model enables parametric studies on various states of contact interfaces. 

It is very important that the magnitude of the second harmonic component be maximized at a specific pressure, and a quantitative relationship between this pressure and the gap needs to be identified in the future. In addition, higher pressure is required to demonstrate this behavior for surfaces that are rougher than those analyzed in this study.

## Figures and Tables

**Figure 1 materials-14-02988-f001:**
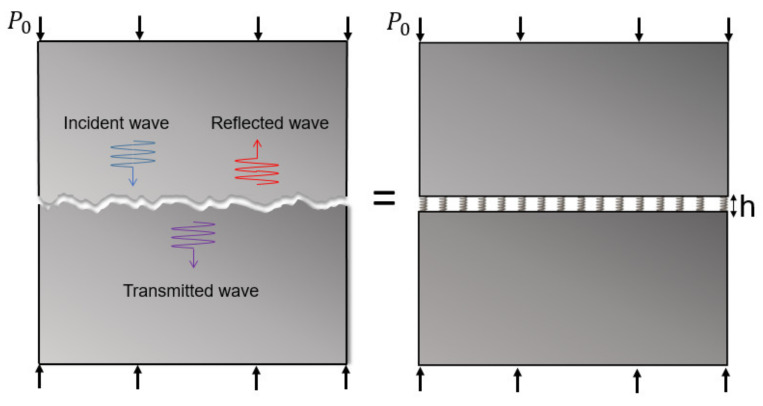
One-dimensional longitudinal plane wave propagation through a rough contact interface and soft contact model of an interface.

**Figure 2 materials-14-02988-f002:**
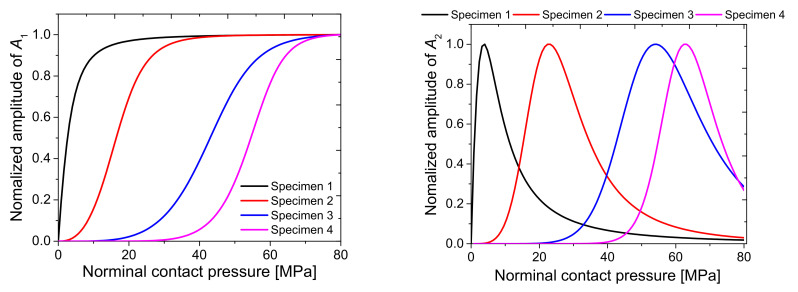
Theoretical simulations of *A*_1_ and *A*_2_.

**Figure 3 materials-14-02988-f003:**
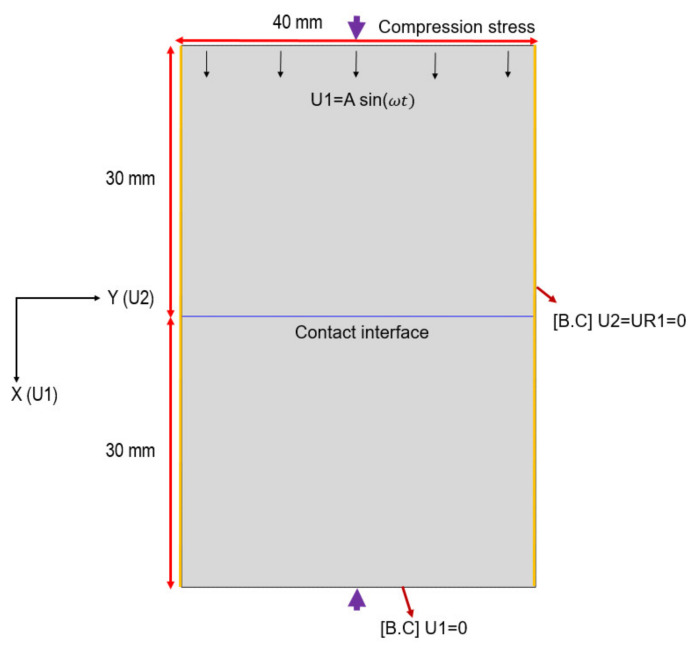
Two-dimensional contact model for FEM analysis.

**Figure 4 materials-14-02988-f004:**
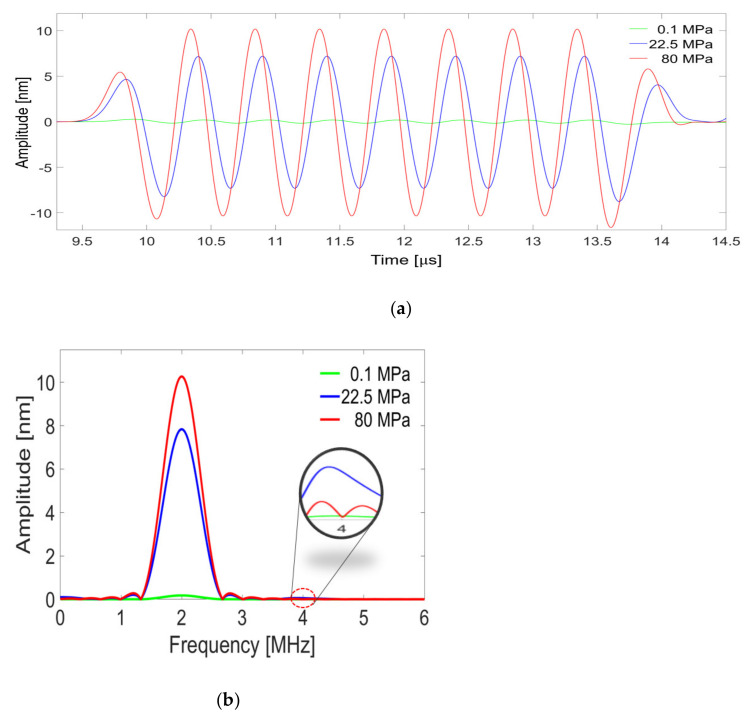
Results of FEM simulation for specimen 2 (C = $4.09\times 10^{-5}$ [${\mathrm{MPa}}^{1-m}\cdot {\mathrm{nm}}^{-1}]$, m = 2.601): (**a**) the received signal and (**b**) the FFT spectrum of the signal in (**a**).

**Figure 5 materials-14-02988-f005:**
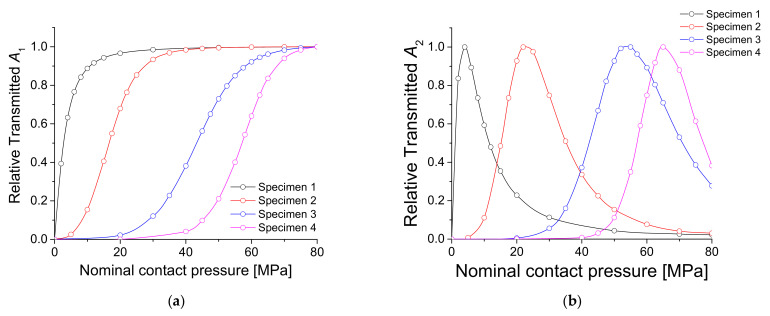
Results of FEM simulations for four specimens: (**a**) *A*_1_ and (**b**) *A*_2_.

**Figure 6 materials-14-02988-f006:**
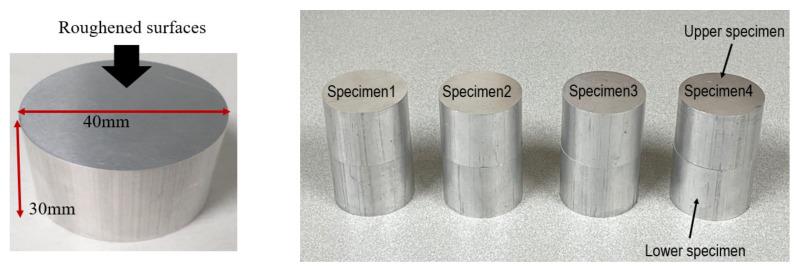
Dimensions of the Al6061-T6 block and four sets of contact interface specimens with different roughness values.

**Figure 7 materials-14-02988-f007:**
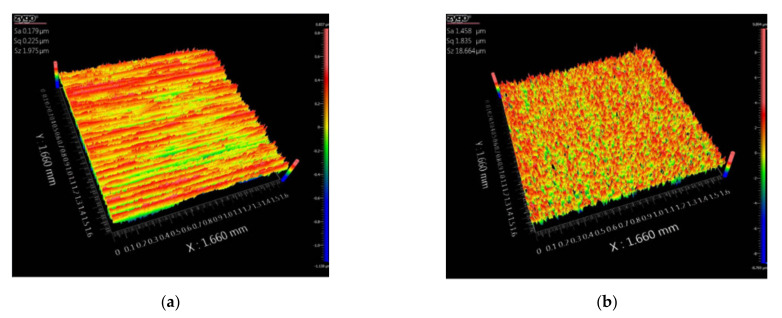
Surface optical microscope images: (**a**) specimen 1, (**b**) specimen 2, (**c**) specimen 3, (**d**) specimen 4.

**Figure 8 materials-14-02988-f008:**
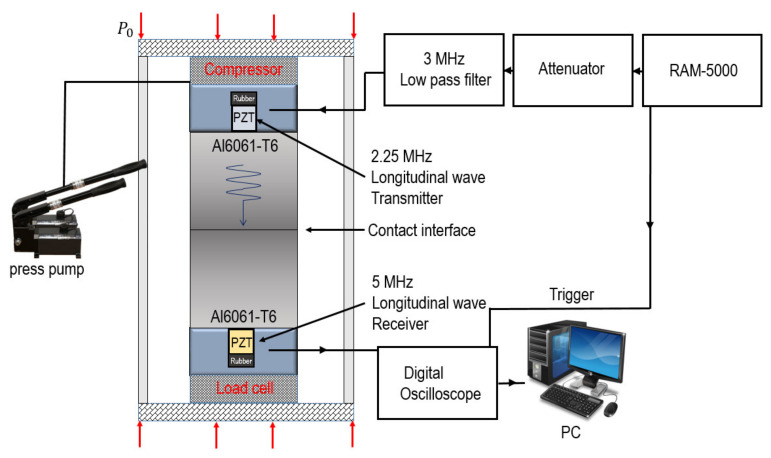
Experimental setup.

**Figure 9 materials-14-02988-f009:**
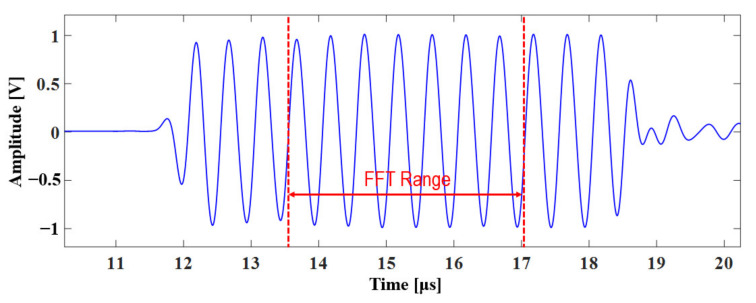
Received signal for specimen 2 at 22.5 MPa.

**Figure 10 materials-14-02988-f010:**
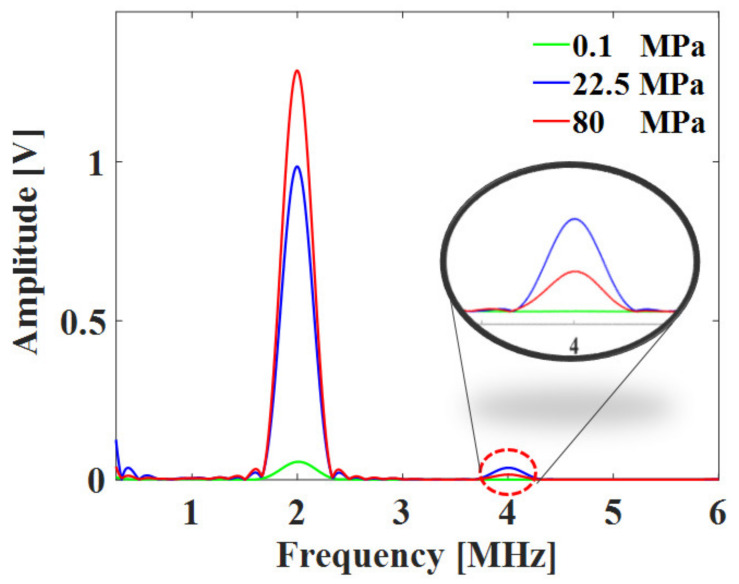
FFT results of received signals for specimen 2 at three contact pressures.

**Figure 11 materials-14-02988-f011:**
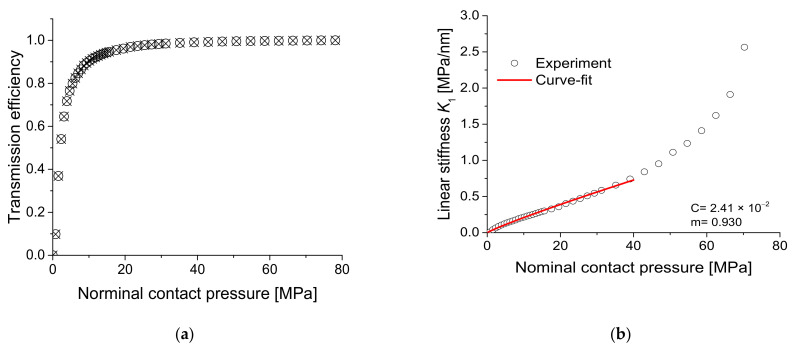
(**a**) Measured transmission efficiency and (**b**) linear stiffness with power-law fitting with respect to contact pressure for specimen 1.

**Figure 12 materials-14-02988-f012:**
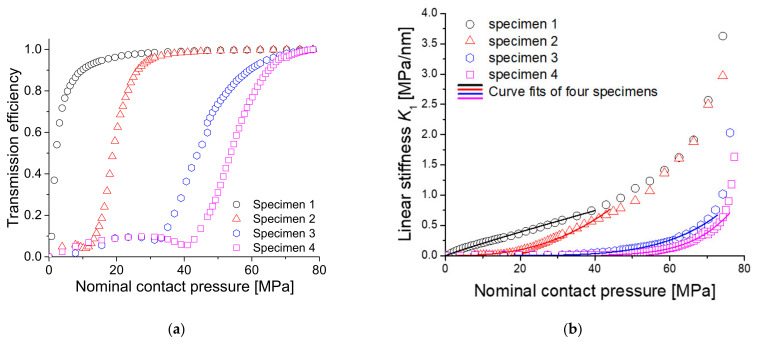
(**a**) Measured transmission efficiency for four specimens and (**b**) linear stiffness with respect to the contact pressure for all specimens with power-law fittings.

**Figure 13 materials-14-02988-f013:**
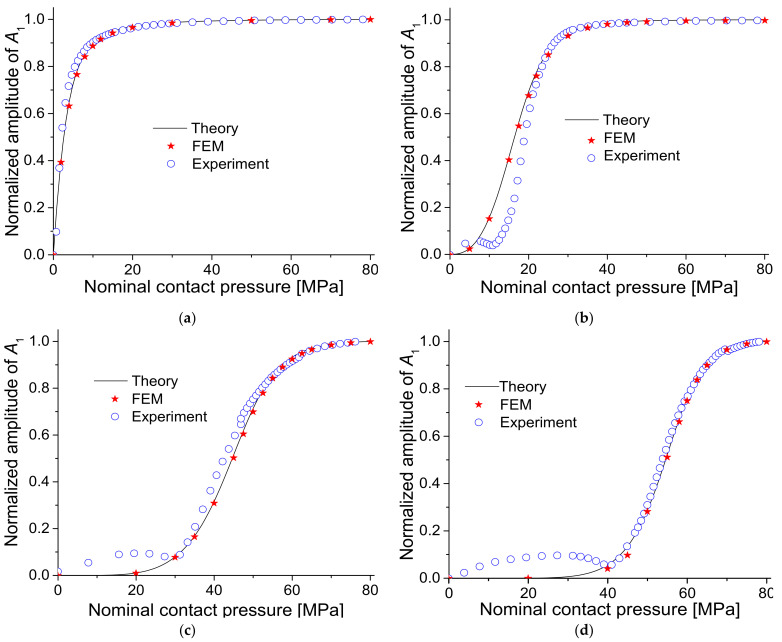
Comparison of theoretical, FEM, and experimental results for *A*_1_: (**a**) specimen 1, (**b**) specimen 2, (**c**) specimen 3, and (**d**) specimen 4.

**Figure 14 materials-14-02988-f014:**
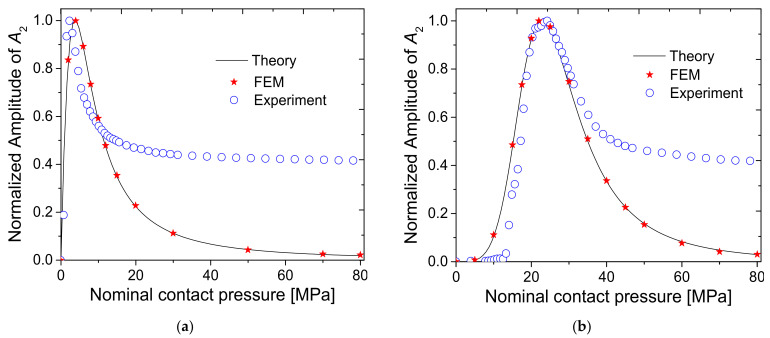
Comparison of theoretical, FEM, and experimental results for *A*_2_: (**a**) specimen 1, (**b**) specimen 2, (**c**) specimen 3, and (**d**) specimen 4.

## Data Availability

The data presented in this study are available on request from the corresponding author.

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
