# Peer review of "Experimental Verification of Contact Acoustic Nonlinearity at Rough Contact Interfaces"

_materials, 2021, doi:10.3390/ma14112988_

Round 1

Reviewer 1 Report

This paper describes an experimental and computational (finite element model) of contact acoustic nonlinearity at rough interfaces. The work is well-done but some work needs to be done on the manuscript before acceptance. The authors show that this tool can be used to measure changes at interfaces which is relevant for finding defects in materials with a broad range of practical applications. However, there are some issues with the discussion of the calculated and theoretical values and the experiment.

Always use the first author’s name with et al. after it when describing work from papers with multiple authors. See p. 2. As an example, use Nam et al. or Biswa et al.

Section 3. Provide more details about the finite element method calcualtions including the number of elements, sized, etc. used in ABAQUS. Also, why are they using the terms used by ABAQUS rather than the physically more relevant terms that they use?

Make sure that the figures are legible and readable as the fonts and symbols are quite small and some are hard to read.

Figure 11. Why was a linear fit used when a linear fit is not always used in figure 12? It looks very non-linear to me. Why are some fits linear and some not in Figures 11 and 12? Also make sure to label Figures as (a) and (b) and correctly in the captions.

In Figure 12, what is happening at high pressure. It looks like there are some very non-linear features.

Give error bars in Tables 2 and 3. Are the values really good to 4 or 5 decimal places? I doubt it.

The explanation for the differences between theory and experiment in Figures 13 and 14 is not satisfactory and they never explain what the differences are due to. They are very significant at low pressure in Figure 13 and at high pressure in Figure 14. Thusm it looks like the theory and FEM results are not at all correct at both high and low pressures depending on what is being examined. They need to explain this much better and how it impacts their hypothesis of using this experimental approach to look at roughened interfaces.

As table 3 is just two numbers, put the numbers for beta prime in the text.

Reviewer 2 Report

The authors of the paper conducted the experimental and numerical investigations to verify the contact acoustic nonlinearity at rough interface between two aluminum blocks. This paper is within the scope of the journal with substantial new model formulation, numerical simulation, test design, and model validation. The paper is well written and can be considered for publication in this journal, provided that the authors can further address the further issues raised by this reviewer:  

  1. In Eq. (2), the variable p should be capitalized P in the two derivatives.
  2. The 2D FEM modeling is based on plane-stress or plane-strain condition?
  3. What is YSYMM? Please give the entire name?
  4. In the test setup of Figure 8, there is stress/strain abrupt change at the region from the PZT area to the loaded area. Does the nonuniform stress distribution disturb the wave propagation in the aluminum blocks? Please further discuss this issue.
  5. The authors need to estimate whether the recorded wave signals include multi-reflections between the interface and the lower boundary according to the wave speed, wave duration, and dimensions?
  6. Line 333, 360, 362, 374: “second harmonic” should be “second harmonic component”?
  7. Line 339: “were the same” should be “was the same”
  8. In Table 3, why not to do determine the beta ratio for the CAN experiment with other pressure values as used in the experiments such as 20, 40, and 60 MPa? Maybe, at the lower pressure, the beta ratio will show difference from that of single block specimen?
  9. The last sentence of Section 5 is not clear. What is a very high pressure? “Verify” should be changed into “demonstrate”?

Round 2

Reviewer 1 Report

The paper is improved but there are still major issues with the work.

There are still too many digits in Table 2 as I noted previously. In addition, the new error bars cannot be correct for C as the error bars are larger than the values. Something is horribly incorrect here. There is a clear issue with the analysis of the data. When the error bars are so large, that means that the values have little if any meaning except as an order of magnitude. This clearly shows that the fits are not correct for C. For m, the error bars are up to 8% of the value so clearly they are not good to 5 or 6 significant digits. The authors need to learn about accuracy.

Make sure that response 7 is in the text, not just in the response letter.

They still need to do a better job of explaining why the FEM and theory deviate from experiment so much in Figure 14 as their explanation is not satisfactory and shows that their modeling and theory approach do not work at higher pressures. What needs to be done in the modeling to improve the agreement? Thus, their results show that their model has only a limited range of validity.

Author Response

This manuscript is a resubmission of an earlier submission. The following is a list of the peer review reports and author responses from that submission.